# Inferring Identity Factors for Grouped Examples

**Shawn Tan, Christopher J. Pal, and Aaron Courville**
MILA, Université de Montréal and École Polytechnique de Montréal
`jing.shan.shawn.tan@umontreal.ca,`
`christopher.pal@polymtl.ca,`
`aaron.courville@umontreal.ca`

## Abstract

We propose a method for modelling groups of face images from the same identity. The model is trained to infer a distribution over the latent space for identity given a small set of "training data". One can then sample images using that latent representation to produce images of the same identity. We demonstrate that the model extracts disentangled factors for identity factors and image-specific vectors. We also perform generative classification over identities to assess its feasibility for few-shot face recognition.

## 1 Introduction and Background

In this article, we propose a method to learn a model to infer a latent representation for an identity, given several examples of images for that identity. This allows us to generate an example for a new concept from a few seen examples, a task Lake et al. (2015) and Rezende et al. (2016) also aim to achieve. Ideally, we would also like the distribution over the identity representation to improve as we observe more examples.

We use Variational Autoencoders (VAEs) (Kingma & Welling, 2013) that are trained to model the image through separate *identity* and *style* latent variables. During training, the model is presented with groups of faces that belong to the same identity, and we demonstrate that the identity and style latent variables control different factors of variation in the image. We also examine the feasibility of using deep generative methods to perform face recognition, and the method also provides us with the benefit of being able to visualise what the model has learnt.

## 2 Method and Related Work

For each image $\mathbf{x}_i$, we assume two associated random variables: $\mathbf{w}$, which contains information related to identity and $\mathbf{z}_i$ which contains image specific factors like pose, lighting, occlusions, etc. The related generative process is as follows,

$$\mathbf{w} \sim p(\mathbf{w}) = \mathcal{N}(\mathbf{0}, \mathbf{I}), \tag{1}$$

$$\mathbf{z}_i \sim p_\theta(\mathbf{z}_i|\mathbf{w}), \qquad \text{for } i = 1, \ldots, N \tag{2}$$

$$\mathbf{x}_i \sim p_\theta(\mathbf{x}_i|\mathbf{z}_i, \mathbf{w}), \qquad \text{for } i = 1, \ldots, N \tag{3}$$

To infer the identity, we design the approximate posterior to be conditioned on all of the images. The per-image latent variable model $\mathbf{z}_i$ is conditioned on the identity, as we believe that knowing the identity can help to disentangle certain underlying factors, for example, knowing a person's skin-tone can affect the inference of lighting factors in the image.

$$\mathbf{w} \sim q_\phi(\mathbf{w}|\mathbf{x}_{1:N}), \tag{4}$$

$$\mathbf{z}_i \sim q_\phi(\mathbf{z}_i|\mathbf{x}_i, \mathbf{w}), \qquad \text{for } i = 1, \ldots, N, \tag{5}$$

Figure 1 illustrates the relationship between the three variables. The resulting loss for this structure

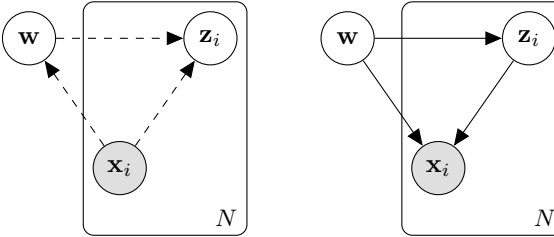

Figure 1: Left: The inference model $q_\phi$ Right: The generative model $p_\theta$. $N$ represents the number of images that are present for each identity.

is as follows,

$$
\begin{aligned}
\log p\left(\mathbf{x}_{1:N}\right) \geq &-\mathbb{E}\left[\log q_\phi\left(\mathbf{w}, \mathbf{z}_{1:N}|\mathbf{x}_{1:N}\right)-\log p\left(\mathbf{w}, \mathbf{z}_{1:N}, \mathbf{x}_{1:N}\right)\right] \\
\approx &\, D_{\mathrm{KL}}\left(q_\phi\left(\mathbf{w}|\mathbf{x}_{1:N}\right)\|p\left(\mathbf{w}\right)\mathbf{x}_{1:N}\right) \\
&+\sum_i^N \mathbb{E}_{q_\phi(\mathbf{w}|\mathbf{x}_{1:N})}\left[D_{\mathrm{KL}}\left(q_\phi\left(\mathbf{z}_i|\mathbf{x}_i, \mathbf{w}\right)\|p_\theta\left(\mathbf{z}_i|\mathbf{w}\right)\right)\right] \\
&-\mathbb{E}_{q_\phi(\mathbf{z}_{1:N},\mathbf{w}|\mathbf{x}_{1:N})}\left[\sum_i^N \log p_\theta\left(\mathbf{x}_i|\mathbf{w}, \mathbf{z}_i\right)\right]
\end{aligned}
$$

If we view the KL-divergence term as a penalty on the information transmitted from the inference model to the generative model, the loss encourages the model to encode as much information as possible in $\mathbf{w}$ as opposed to in $\mathbf{z}_i$, which is penalised as many times as there are images.

Such a structure for conditioning on an entire set of data to approximate posterior distribution over a common latent variable has been proposed in Edwards & Storkey (2016), with an experiment on face image data extracted from Youtube videos. Here, however, we make the simplifying assumption that the style latent variables are independent given the identity. In speech, Tan & Sim (2016) uses the same technique to extract a latent representation for an utterance, which works well for robust speech recognition. Hsu et al. (2017) also proposed a similar graphical model structure, using an RNN for their encoder and decoder.

### 2.1 GENERATIVE FACE RECOGNITION

Given a set of images for a new identity, we can generate a new image with the same identity. We use the inference model $q_\phi\left(\mathbf{w}|\mathbf{x}_{1:N}\right)$ to obtain an estimate of $\mathbf{w}$ by sampling. We can then use ancestral sampling to obtain an image under the new identity. We can use the same process to estimate the probability a given image given an identity vector $\mathbf{w}$, using importance sampling:

$$
p\left(\mathbf{x}|\mathbf{w}\right)=\mathbb{E}_{q_\phi(\mathbf{z}|\mathbf{w},\mathbf{x})}\left[\frac{p_\theta\left(\mathbf{x}|\mathbf{z}, \mathbf{w}\right)p_\theta\left(\mathbf{z}|\mathbf{w}\right)}{q_\phi\left(\mathbf{z}|\mathbf{w}, \mathbf{x}\right)}\right] \tag{6}
$$

$$
\approx \frac{1}{K}\sum_{k=1}^K \frac{p_\theta\left(\mathbf{x}|\mathbf{z}_{(k)}, \mathbf{w}\right)p_\theta\left(\mathbf{z}_{(k)}|\mathbf{w}\right)}{q_\phi\left(\mathbf{z}_{(k)}|\mathbf{w}, \mathbf{x}\right)} \qquad \mathbf{z}_{(1)}, \ldots, \mathbf{z}_{(K)} \sim q_\phi\left(\mathbf{z}|\mathbf{w}, \mathbf{x}\right) \tag{7}
$$

We then define the probability of $\mathbf{w}$ given $\mathbf{x}$ as,

$$
\tilde{p}\left(\mathbf{w}|\mathbf{x}\right) \triangleq \frac{p\left(\mathbf{x}|\mathbf{w}\right)}{\sum_{\mathbf{w}'} p\left(\mathbf{x}|\mathbf{w}'\right)} \tag{8}
$$

## 3 EXPERIMENTS

For our experiments, $\phi$ and $\theta$ are parameterised based on the architecture used in Tan & Sim (2016). $q_\phi\left(\mathbf{w}|\mathbf{x}_{1:N}\right)$ and $q_\phi\left(\mathbf{z}|\mathbf{w}, \mathbf{x}\right)$ have a shared convolution and max pooling transforms that we will call $f(\mathbf{x}_i)$. For $q_\phi\left(\mathbf{w}|\mathbf{x}_{1:N}\right)$, we perform a mean-pooling operation over the images in the dataset

before passing it through an MLP to output the mean and standard deviation for $\mathbf{w}$. $q_\phi(\mathbf{z}|\mathbf{w}, \mathbf{x})$ is also an MLP that takes in $\mathbf{w}$ and the output of $f(\mathbf{x}_i)$. For $\theta$, $p_\theta(\mathbf{z}_i|\mathbf{w})$ is also an MLP. $p_\theta(\mathbf{x}_i|\mathbf{z}_i, \mathbf{w})$ is parameterised by a stack of upsampling and convolutional transforms, and we perform conditonal instance normalisation on $\mathbf{z}_i$ at each layer, in an architecture similar to the one presented in Dumoulin et al. (2016). The architecture was implemented in Theano (Team et al., 2016).

**Learnt representation space**    To demonstrate that $\mathbf{z}_i$ and $\mathbf{w}$ encode different factors of variability in an image, we extract $\mathbf{w}_1$ from Identity 1, and then extracting from two images $\mathbf{z}_{1,1}$ and $\mathbf{z}_{1,2}$ we perform linear interpolation between these points. We then generate 10 images using the sequence of $\mathbf{z}$ (10 co-linear points) with $\mathbf{w}_1$ (first row Figure 2c) and $\mathbf{w}_2$ (second row Figure 2c). We see that traversing the $\mathbf{z}$ linearly for both identities results in the same pose and background change, with a different face.

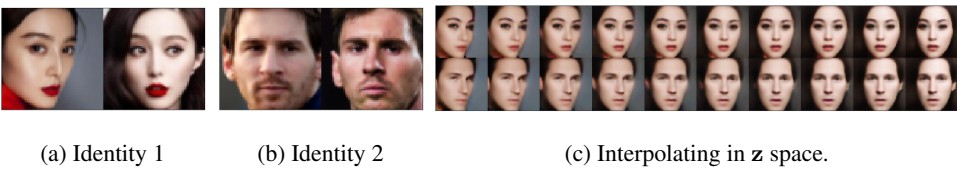

(a) Identity 1          (b) Identity 2                    (c) Interpolating in $\mathbf{z}$ space.

Figure 2

**Face recognition**    We trained the model on the MSCeleb dataset, and performed face recognition on a held-out set of 2000 identities from the training set. We vary the number of examples ($N$) used to infer $\mathbf{w}$, and using the method in Section 2.1 to predict across the 2000 identities on an unseen image. The instances to accuracy trade-off can be seen in Figure 3a. Unfortunately, even with 20 seen examples, the model only performs just over 50% on a held-out testing image, but it is interesting to note that, while only training with $N = 10$, the model still shows an improvement even as we increase $N$ beyond that. We can also visualise when $\mathbf{z}_i = \boldsymbol{\mu}_{\mathbf{z}_i}$ looks like, as seen in Figure 3b.

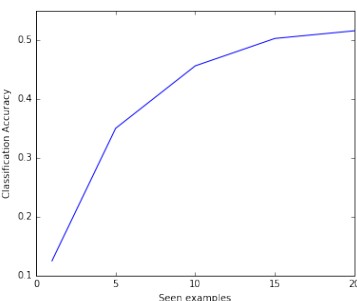
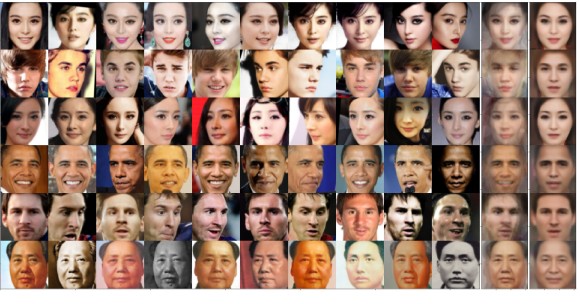

(a) As the number of seen instances ($N$) increases, the accuracy of recognition improves.

(b) Left: Seen examples from the MSCeleb dataset. Each row represents a group of images from one identity. 2nd column from right: The mean of the seen examples. Far right: Visualising $\mathbf{x}_i$ when $\mathbf{z}_i = \boldsymbol{\mu}_{\mathbf{z}_i}$.

Figure 3

## 4   CONCLUSION AND FUTURE WORK

We believe that few-shot generalisation of a new concept is an important capability for machine learning systems and being able to recognise a face after seeing just a few instances is one realisation of this. It is also important enough to be able to have such face recognition be interpretable and diagnosable. Interpretability will prove useful in debugging problems with facial recognition system, especially as they become more prevalent with their introduction into everyday devices. We therefore think this is a worthwhile direction, with many possibilities for future work.

ACKNOWLEDGMENTS

We thank Huang Chin-wei, Shen Yikang, Francis Dutil, and Joseph Paul Cohen for their helpful and insightful comments during the course of this project.

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
