# OpenReview forum: "Inferring Identity Factors for Grouped Examples"
_ICLR.cc/2018/Workshop — Reject_

### Official Review · AnonReviewer1 · 2018-02-22
**Limited novelty**

**Rating:** 4
**Confidence:** 3

**Review:**

This paper describes a variational framework that attempts to disentangle face identity from 'style' factors (where 'style' corresponds to all the other factors of variation).  The authors' goal is to build a system that can generate samples of a novel concept in a few-shot inference regime, where the identity of the new concept is inferred from few example images of that concept (similarly to Lake et al, 2015 and Rezende et al, 2016).

Pros:
- The abstract is well written
- The limited samples presented in the abstract look decent
- It appears that the authors were able to disentangle facial identity from the rest of the generative factors

Cons:
- The model is not novel, as the authors admit themselves.
- The authors are not able to achieve the few-shot inference objective they were aiming at

Due to the limited novelty and initial results that do not look as promising as even the authors would have liked, I am struggling to find a reason to accept this paper.

---

### Official Review · AnonReviewer3 · 2018-03-13
**Interesting paper inferring grouped factors for disentanglement, but missing links to important prior work**

**Rating:** 3
**Confidence:** 5

**Review:**

This paper proposed to learn disentangled factors of variation by learning from groupings of samples and recognizing that latent factors can be separated into different levels of locality by clamping certain variables across samples. This leads to more sample-efficient learning as information is shared, as has been recognized in many previous works.

This work unfortunately misses the important reference to "Multi-Level Variational Autoencoder: Learning Disentangled Representations from Grouped Observations" by Bouchacourt et. al (AAAI 18) which has tackled the same problem with a more general treatment based on the same idea.

I would advise that the authors clarify the relationship to that work to improve their exposition.

---

### Decision · Program_Chairs · 2018-03-20
**ICLR 2018 Workshop Acceptance Decision**

**Decision:**

Reject

**Comment:**

Based on the reviews, this paper has not been accepted for presentation at the ICLR workshop. However, the conversation and updates can continue to appear here on OpenReview.